# [RE] Nondeterminism and Instability in Neural Network Optimization

## Reproducibility Summary

### Scope of Reproducibility

The claims of the paper [1] are threefold: (1) Summers and Dinneen [1] made the surprising yet intriguing discovery that all sources of nondeterminism exhibit a similar degree of variability in the model performance of a neural network throughout the training process. (2) To explain this fact, they have identified model instability during training as the key factor contributing to this phenomenon. (3) They have also proposed two approaches (Accelerated Ensembling [2] and Test-Time Data Augmentation [3]) to mitigate the impact on run-to-run variability without incurring additional training costs. In the paper [1], the experiments were performed on two types of datasets (image classification and language modelling). However, due to the intensive training and time required for each experiment, we will only consider image classification for testing all three claims.

### Methodology

Our approach to investigating the claims made in the paper [1] can be divided into three parts: (1) Replication: we used the publicly available code and adapted it to our experimental environment with some modifications to replicate the results; (2) Ablation study: we tried to use different parameters, reducing the total implementation time to less than half compared to the original study, while keeping the central claim intact; (3) Generalization: we studied the authors' claim on a much more complex dataset and architecture to gain insights on the reproducibility of the conclusion. All experiments necessarily required extensive training, with a single experiment alone requiring 490 hours of 2 Nvidia Tesla V100 16GB (i.e., 700 trained models).

### Results

With our tests and the obtained results, we confirm that all individual and combined sources of nondeterminism have similar effects on model variability and that instability in neural network optimization is the main reason for this phenomenon. However, our results show some discrepancies in the reduction of variability by test-time data augmentation (TTA) and accelerated ensembling (claim 3 above). Like the original study, we show that these approaches successfully reduce variability, but the degree of reduction is reported as $61\%$, whereas our study reports $51\%$ as the highest value. Despite some small differences, the third claim remains and we support it.

### What was easy

The authors have made the source code publicly available in the GitLab repository. Even without extensive documentation, reimplementation of the experiments was straightforward and required little effort. Moreover, the paper's clearly presented details significantly reduced the effort required to set up the experimental configurations. The use of regular neural network training and widely used datasets was the icing on the cake to follow the implementation. This allowed us to explore other new aspects of the method.

Submitted to ML Reproducibility Challenge 2021. Do not distribute.

**What was difficult**

Although the implementation was easy to comprehend and intuitive with the resources provided, the validation of some baselines proved to be computationally intensive and time-consuming, requiring multiple runs. In particular, the variability analysis required training 100 models each for 500 epochs to verify the role of a single source of nondeterminism. Nevertheless, we managed to maintain the original settings, but we could not run multiple iterations to gain more confidence in the results.

**Communication with original authors**

At the beginning of our reproducibility study, we contacted the original authors once. The basic questions about the experimental settings were answered and the foundation for the rest of our experiments was laid. In addition, we also referred to their post and answers available on the OpenReview portal.

# 1 Introduction

In the pursuit of reproducibility in Deep Learning, a key criterion is the elimination of sources of nondeterminism in model optimization. Random initialization of weights is considered the main source of nondeterminism [4], but other sources such as random shuffling of training data [5], random data augmentation [6], and even GPU libraries such as cuDNN also contribute [7] . These random parameters tend to initialize with a random value every time we train the model, even if we use the same source code. On the one hand, such randomization helps to achieve sound performance, but on the other hand, it leads to run-to-run variability. This causes difficulties in verifying and improving baselines. To have complete experimental control, a better understanding of these random components is required, which is why each independent model is trained multiple times as a standard practice. While this can solve the problem, it is extremely costly in terms of computational resources and time.

The authors' original work focused on quantifying the independent effect of each source of nondeterminism on model training. All different sources of nondeterminism were found to have similar effects on model variability. They also created an experimental protocol that used standard evaluation measures of model diversity and variability to capture model behaviour better. While developing a basic mechanism of understanding, they also discovered model instability as a major cause of run-to-run variability. To support this finding, experiments were conducted on image classification and language modeling datasets. In the end, two solutions were proposed to reduce variability without additional costs.

# 2 Scope of reproducibility

1) First, we have attempted to replicate all three of the paper's claims:

- Claim 1: All sources of nondeterminism have similar effects on model diversity and variability.

This claim seems to be a surprising discovery, as it could pave the way for researchers to improve the algorithm as a whole to reduce the effects of model variability, rather than focusing on each source of nondeterminism separately. In this reproducibility report, all sources of nondeterminism were tested individually and also in combination with other sources for ResNet-14 [8] on the CIFAR-10 dataset [9].

- Claim 2: The key driver of this phenomenon "All sources of non-determinism have similar effects on model diversity and variability" is the instability of model optimization.

Model optimization is said to have instability where small changes to the initial parameters lead to large changes to the final parameter values. Simply put, changing the initialization of a single weight by the smallest possible amount of $10^{-10}$ has the same effect as initializing all weights with completely random values. This study shows that any source of nondeterminism is susceptible to a change in weights by at least $10^{-10}$ and therefore produces the same amount of variability. This also illuminates the discovery of Zhuang et al. [4], which shows that removing a single source of nondeterminism is not sufficient to improve the stability of the training.

- Claim 3: Accelerated ensembling and TTA are two possible solutions to reduce model variability without additional training costs (e.g., time).

As mentioned earlier, the standard practice to counter model variability is to train models multiple times, which costs additional computational resources. This claim attracts our attention because it could change current practices and facilitate the reproducibility of experiments in the context of Deep Learning by promoting deterministic training without additional costs.

2) Although the main objective of our study is to reproduce the main claims, we did not limit ourselves to these only. We have also conducted a series of experiments that go beyond the paper and follow two lines of investigation:

- Ablation study: During the reproducibility study, one of the main difficulties we faced was the excessive amount of time required to conduct a single experiment. In Section 4.2, we attempted to address this issue by recommending changes to the default settings supported by our experimental results, while keeping the main claims intact.

- Generalization to larger architectures and datasets: Because we were able to reduce the experimental time, we tested the authors' claim on a larger architecture and dataset to verify that the claim still holds in general.

## 3   Methodology

### 3.1   Code

The publicly available source code was provided by the original authors in the corresponding GitHub repository . It was written using the Pytorch [10] and NumPy libraries with Python 3.7.5. We used the same code and made some adjustments, such as setting the width of the output terminal and downloading the datasets manually. With these minor changes, we were able to run the code according to our experimental environment. The code provides a basic structure that allows numerous architectures to be used with few changes. However, additional code must be added for architectures larger than ResNet-18 [8]. The code also has command line functionality that allows the user to configure seed values and hyperparameter settings for a specific task. Although the code covers the entire implementation with the exact experimental settings described in the paper, a portion of the code is missing to visualize the results.

### 3.2   Model descriptions

We initially chose the ResNet-14 model because it was frequently used for image classification experiments in the original study along with the CIFAR-10 datasets. Due to the large number of trained models required for a single experiment, we did not have time to work with other models except ResNet-34, which we created ourselves to test the generalization of the claim over larger model architectures. Figure 1 shows the basic residual block of a ResNet architecture, consisting of two 3x3 convolutional layers followed by batch normalization before activation. On the same basis, Table 1 shows the modular architecture of ResNet-14 and ResNet-34 [8] without the first 7x7 convolutional layer and the final fully-connected layer. The blocks are shown in parentheses along with the number of output channels, with the multiplier indicating the number of residual blocks in that module.

| Module No. | ResNet-14 | ResNet-34 |
|:---:|:---:|:---:|
| 1 | $\begin{bmatrix} 3 \times 3, & 16 \\ 3 \times 3, & 16 \end{bmatrix} \times 2$ | $\begin{bmatrix} 3 \times 3, & 64 \\ 3 \times 3, & 64 \end{bmatrix} \times 3$ |
| 2 | $\begin{bmatrix} 3 \times 3, & 32 \\ 3 \times 3, & 32 \end{bmatrix} \times 2$ | $\begin{bmatrix} 3 \times 3, & 128 \\ 3 \times 3, & 128 \end{bmatrix} \times 4$ |
| 3 | $\begin{bmatrix} 3 \times 3, & 64 \\ 3 \times 3, & 64 \end{bmatrix} \times 2$ | $\begin{bmatrix} 3 \times 3, & 256 \\ 3 \times 3, & 256 \end{bmatrix} \times 6$ |
| 4 | | $\begin{bmatrix} 3 \times 3, & 512 \\ 3 \times 3, & 512 \end{bmatrix} \times 3$ |

Table 1: Architecture of ResNet-14 and ResNet-34 in modular format

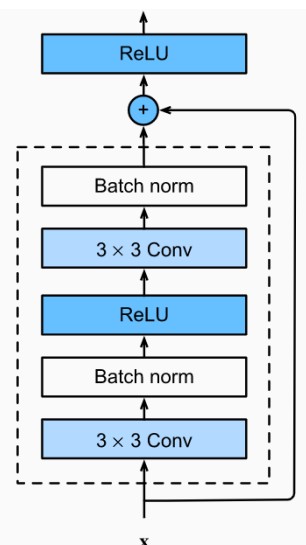

Figure 1: ResNet residual block

## 3.3   Datasets

To investigate the effects of nondeterminism on image classification, the authors used CIFAR-10 as the primary training dataset to train the ResNet-14 model. Because the architectures used in the original study were smaller, the use of CIFAR-100 was not seen. In comparison, we additionally used the CIFAR-100 dataset to train and test the larger architecture (ResNet-34) in our extended study. It is worth noting that the authors used a runtime code to download the datasets using the torchvision library. Although this would have been the preferred choice, we encountered some issues due to the external access limitations of our experimental environment. Thus, we manually downloaded the dataset and specified the path, which worked well for us. We used the authors' protocol for the training and testing portions with $50k$ samples and $10k$ samples of CIFAR-10, respectively. No validation set was used. Table 2 shows the datasets used in our experimental tests.

| Dataset | Classes | Samples | Dimensions | Split(train/val/test) |
|---|---|---|---|---|
| CIFAR-10 | 10 | 60k | $32 \times 32 \times 3$ | 50k/0/10K |
| CIFAR-100 | 100 | 60k | $32 \times 32 \times 3$ | 50k/0/10K |

Table 2: Summay of datasets

## 3.4   Hyperparameters

In training the models, all standard parameters were used as given in the paper to more closely approximate the original approach. While not all values were mentioned in the paper, they could be easily found in the code itself. All models were trained with a cosine learning rate decay [11] with a maximum learning rate of $0.40$ and $500$ epochs. We also used the first three epochs for warm-up with linear learning rate, as was the case in the original study. Throughout the training, the SGD optimizer was used with a batch size of $512$, a momentum of $0.9$, and a weight decay of $5.10^{-4}$.

## 3.5   Evaluation for the effects of nondeterminism

To understand the impact of each source of nondeterminism, the original study developed a protocol related to performance variability and model diversity representation.

**Performance variability:** All random sources are controlled by seeding values. To test a single source, all other sources are assigned the constant deterministic seeding value of $1$, except for the source that is under observation and assumes different seeding values from $1$ to the total number of training runs. For the sources that cannot be seeded, such

as cuDNN, the range is limited to 0 to 1, with 0 and 1 indicating the random and deterministic values, respectively. In the original settings, the total number of training runs is set to 100, so for each source of nondeterminism, 100 models can be trained. If we assume 4 different sources, each can be represented as $S1, S2, S3$, and $S4$, where $S$ denotes the seed values. For example, $S1$ denotes the seed for random parameter initialization, $S2$ for training data shuffling, $S3$ for data augmentation, and $S4$ for cuDNN. If we set $(S1 = 1, S2 = 1, S3 = R, S4 = 1)$, where $R$ denotes the range from 1 to the total number of training runs, we would analyse the effect of data augmentation as a random source. If we want to analyse the effects of multiple sources simultaneously, we would also assign the $R$ values to other sources. Finally, to work with performance variability, we consider the standard deviation of accuracy and cross entropy across all 100 trained models for each source.

**Representation diversity:** In addition to performance variability, the authors also considered the representation of the trained models. This allows us to determine the difference in the representation of the trained models even when their performance variability is the same. In doing so, they used four different metrics that we followed. Of these, we did not find an implementation for the Centered Kernel Alignment (CKA) evaluation metric [12], which is considered the most advanced evaluation metric for determining similarities between models. The first metric is the simplest, which uses the average disagreement between pairs of 100 models. Second, they used the average correlation between the models' predictions. Finally, they examined the change in performance when two models are ensembled from the same source of nondeterminism.

## 3.6 Extended experiments

With the extended experiments, we have tried to eliminate the difficulties we encountered in replicating the experiments to verify the paper's claim. One of our main concerns was the time required for each experiment. Therefore, we performed a series of experiments beyond the original work to satisfy the claims without being computationally intensive.

**Models v/s model variability:** In this experiment, we examine the actual number of models required to test the variability analysis for each source of nondeterminism, which can reflect the same conclusion as the original settings while reducing the overall computational cost. Due to time constraints, we chose to work with only two sources. We varied the number of models for each source and observed the results using the same evaluation measures.

**Epochs v/s model variability:** Another factor contributing to the long training time is the use of a large number of epochs. In this experiment, we investigate the effects of a different number of epochs on the variability of the model, and therefore try to obtain similar results with a smaller number of epochs.

## 3.7 Experimental setup and computational requirements

To achieve similar results as in the original study, we strictly follow the same experimental environment and use Pytorch as framework with Python = 3.7.5, NumPy = 1.17.4, Torch = 1.3.1 and Torchvision = 0.4.2. All experiments were performed on the HPC cluster ARA using the SLURM workload manager at Friedrich Schiller University Jena. This system consists of multicore nodes for high computational performance and therefore offers a variety of GPU systems that can be used. According to our needs, we chose to work with 2 NVIDIA Tesla V100 GPUs equipped with 24 core Intel CPU and 128 GB of RAM. Table 3 shows the number of experiments performed and the time needed for them. It is worth mentioning that the evaluation part of all experiments involves the technique of TTA, which must also be performed on a GPU, so the time needed for the evaluation part is also added. The first three experiments belong to each of the three claims and the others are part of our extended study, which is not included in the original studies.

| Experiments | Dataset | Model | Total No. of trained models | Epochs | Total training time (hrs) | Total evaluation time (hrs) |
|---|---|---|---|---|---|---|
| Exp-1 | CIFAR-10 | ResNet-14 | 700 | 500 | 490 | 10.5 |
| Exp-2 | CIFAR-10 | ResNet-14 | 100 | 500 | 70 | 1.5 |
| Exp-3 | CIFAR-10 | ResNet-14 | 100 | 500 | 75 | 1.5 |
| Ext.Exp-1 | CIFAR-10 | ResNet-14 | (10,25, 50,100) | 500 | 140 | 2.5 |
| Ext.Exp-2 | CIFAR-10 | ResNet-14 | 125 | (100 to 500) | 90 | 1.7 |
| Ext.Exp-3 | CIFAR-100 | ResNet-34 | 25 | 200 | 75 | 2.5 |

Table 3: Time required for each experiment

# 4   Results

In this section we present our experimental results by replicating all three claims and going beyond the paper [1]. First, we start with the core replication by following exactly in the footsteps of the original authors and producing all three experiments for each claim. Second, we ran three more experiments that help us obtain the same results much faster and generalize the first two claims across different datasets and architectures.

## 4.1   Core replication results

**Effects of nondeterminism sources**   Table 4 shows the result of our replication study for claim 1 with some minor differences as reported in [1]. In addition to all sources of nondeterminism, we performed additional deterministic training to verify that we have complete control over all sources and no other effects of randomness are observed during the training process unless otherwise noted. We trained all 100 models for each source, with 100 different seed values, as in the original work. We obtained almost the same results when we analysed the effect of each source separately. However, when the combination of multiple sources was tested, we found only a few anomalies (marked with red colour in Table 4), which ultimately appear to be negligible. Therefore, we support the claim that all sources of nondeterminism have similar effects on model variability and diversity.

| Nondeterminism Sources | Accuracy SD (%) | Cross-Entropy SD (%) | Pairwise Disagree (%) | Pairwise Corr. | Ensemble $\Delta$(%) |
|---|---|---|---|---|---|
| Determinism | 0 | 0 | 0 | 1 | 0 |
| Parameter Initialization | $0.22 \pm 0.02$ | $0.0073 \pm 0.0005$ | 10.7 | 0.873 | 1.86 |
| Data Shuffling | $0.25 \pm 0.02$ | $0.0083 \pm 0.0005$ | 10.7 | 0.871 | 1.86 |
| Data Augmentation | $0.22 \pm 0.02$ | $0.0069 \pm 0.0005$ | 10.7 | 0.872 | 1.87 |
| cuDNN | $0.21 \pm 0.01$ | $0.0067 \pm 0.0004$ | 10.6 | 0.873 | 1.83 |
| Data Shuffling + cuDNN | **0.25** $\pm 0.02$ | $0.0071 \pm 0.0005$ | 10.7 | 0.871 | 1.85 |
| Data Shuffling + Aug + cuDNN | **0.24** $\pm 0.02$ | $0.0068 \pm 0.0004$ | 10.7 | 0.871 | 1.87 |
| All Nondeterminism Sources | $0.26 \pm 0.02$ | $0.0080 \pm 0.0005$ | 10.7 | 0.871 | 1.84 |

Table 4: Effects of nondeterminism sources

**The effect of instability**   Table 5 shows the result of the second claim. Note that the second claim states that instability in neural network optimization is the key factor for similar effects of nondeterminism sources on model variability. To observe the effects of instability, deterministic training was performed with a small change of 1 bit ($5 \times 10^{-10}$) in a single random weight for 100 models. Our results show that this 1-bit change generates about as much variability as any other source of nondeterminism. Therefore, this claim can be considered confirmed by our experiment.

| Nondeterminism Sources | Accuracy SD (%) | Cross-Entropy SD (%) | Pairwise Disagree (%) | Pairwise Corr. | Ensemble $\Delta$(%) |
|---|---|---|---|---|---|
| Random Bit Change | $0.21 \pm 0.014$ | $0.0067 \pm 0.0004$ | 10.6 | 0.874 | 1.82 |

Table 5: Effects of instability

**Reduction of variability with proposed methods**   In Table 7 we show the results of our replication study for the third claim. Unlike the other two claims, our study this time shows numerous differences in the results compared to the original study. The TTA and accelerated ensembling methods were used to reduce model variability, and the values were compared to the result obtained by combining all sources of nondeterminism presented as a "Single Model". First, we found a computational error in the percentage of variability reduction in the original paper itself, which was taken as the average value for all 5 metrics compared to the single model. This measure would allow us to see the effectiveness of these two probable solutions. Although the mathematical formula is not given in the study, it seems intuitive to work with average values. Therefore, we performed a calculation of the baseline averages and found differences in the overall reduction percentages. For all reduction percentages, the values on paper appear to be about 20% higher than the calculated values. This is highlighted in red, as shown in Table 6.

Second, in addition to the calculation error, we found other minor anomalies that account for more than 10% change, as shown in Table 7 (highlighted in red). Including all minor differences, we have again shown that TTA and accelerated ensembling can be used to reduce variability. However, the highest possible percent reduction was reduced from 61% to 51% compared to the original study. When compared to the recalculated values, a slight increase in the percentage is observed. Moreover, the different types of TTA alone seem to cause an equal reduction in performance variability, while the change is mainly visible in model diversity. Thus, the overall variability is reduced. Despite these differences, the third claim remains and we therefore support it..

| Nondeterminism Sources | Training Cost | Accuracy SD (%) | Cross-Entropy SD (%) | Pairwise Disagree (%) | Pairwise Corr. | Ensemble $\Delta$(%) | Variability Reduction (%) |
|---|---|---|---|---|---|---|---|
| Single Model | 1x | $0.26 \pm 0.02$ | $0.0072 \pm 0.0005$ | 10.7 | 0.871 | 1.82 | n/a |
| Acc.Ens. | 1x | $0.19 \pm 0.02$ | $0.0044 \pm 0.0003$ | 6.1 | 0.957 | 0.63 | ~~48~~37 |
| Single/Flip-TTA | 1x | $0.24 \pm 0.02$ | $0.0061 \pm 0.0005$ | 8.2 | 0.905 | 1.20 | ~~21~~17 |
| Single/Crop25-TTA | 1x | $0.23 \pm 0.02$ | $0.0059 \pm 0.0004$ | 9.2 | 0.893 | 1.49 | ~~16~~13 |
| Single/Crop81-TTA | 1x | $0.21 \pm 0.01$ | $0.0055 \pm 0.0004$ | 8.8 | 0.898 | 1.39 | ~~21~~17 |
| Single/Flip-Crop25-TTA | 1x | $0.21 \pm 0.02$ | $0.0051 \pm 0.0004$ | 7.2 | 0.920 | 0.99 | ~~33~~26 |
| Single/Flip-Crop81-TTA | 1x | $0.19 \pm 0.01$ | $0.0049 \pm 0.0004$ | 6.9 | 0.922 | 0.92 | ~~37~~30 |
| Acc.Ens/Flip-TTA | 1x | $0.15 \pm 0.01$ | $0.0039 \pm 0.0003$ | 5 | 0.967 | 0.45 | ~~58~~46 |
| Acc.Ens/Flip-Crop81-TTA | 1x | $0.16 \pm 0.01$ | $0.0033 \pm 0.0002$ | 4.6 | 0.972 | 0.38 | ~~61~~48 |

Table 6: Correction in percentage reduction

| Nondeterminism Sources | Training Cost | Accuracy SD (%) | Cross-Entropy SD (%) | Pairwise Disagree (%) | Pairwise Corr. | Ensemble $\Delta$(%) | Variability Reduction (%) |
|---|---|---|---|---|---|---|---|
| Single Model | 1x | $0.26 \pm 0.02$ | $\mathbf{0.0080} \pm 0.0005$ | 10.7 | 0.871 | 1.84 | n/a |
| Acc.Ens. | 1x | $\mathbf{0.17} \pm 0.01$ | $0.0043 \pm 0.0003$ | 6.1 | 0.957 | 0.65 | 40 |
| Single/Flip-TTA | 1x | $0.22 \pm 0.01$ | $0.0063 \pm 0.0004$ | 8.0 | 0.905 | 1.17 | **20** |
| Single/Crop25-TTA | 1x | $0.22 \pm 0.02$ | $0.0057 \pm 0.0004$ | 9.1 | 0.893 | 1.46 | **17** |
| Single/Crop81-TTA | 1x | $0.22 \pm 0.01$ | $0.0056 \pm 0.0004$ | 8.7 | 0.897 | 1.36 | **19** |
| Single/Flip-Crop25-TTA | 1x | $0.22 \pm 0.01$ | $0.0050 \pm 0.0003$ | 7.1 | 0.919 | 0.98 | 28 |
| Single/Flip-Crop81-TTA | 1x | $\mathbf{0.21} \pm 0.01$ | $0.0050 \pm 0.0004$ | 6.9 | 0.922 | 0.92 | 30 |
| Acc.Ens/Flip-TTA | 1x | $0.16 \pm 0.01$ | $0.0037 \pm 0.0002$ | 5 | 0.967 | 0.42 | 47 |
| Acc.Ens/Flip-Crop81-TTA | 1x | $0.15 \pm 0.01$ | $0.0031 \pm 0.0002$ | 4.5 | 0.972 | 0.37 | 51 |

Table 7: Reproducibility study for variability reduction. Prominent differences accounted for morethan 10% are shown in red.

## 4.2 Additional results not present in the original paper

In replicating all three claims, we faced the major problem of the time required to produce the results. While it is understandable that the nature of the problem requires multiple model training, it is still not clear to us why the author

206  used 100 trained models with 500 epochs each as default settings for each source of nondeterminism. Since these two
207  parameters play an important role in the time required for an experiment, we decided to explore this area with the goal
208  of reducing the training time while maintaining all the claims. This will allow the scientific community to test the
209  claims on larger architectures and datasets with less time consumption.

210  **No. of models v/s Variability**   In this experiment, we used the original settings, except for the number of trained
211  models considered for the variability analyses.  By changing the number of trained models for nondeterminism,
212  we observed the change in model variability.  Due to lack of time, we experiment with only 3 different sources of
213  nondeterminism. We found that the result is not significantly different from the number of models, except for the error
214  bars associated with standard deviation of accuracy and cross entropy. While all important metrics remain the same, it
215  can be observed that these error bars decrease as the total number of models increases, as can be seen in Figure 2. All
sources tested in Table 8 show the same trend.

| Setting | Accuracy SD (%) | Cross-Entropy SD (%) | Pairwise Disagree (%) | Pairwise Corr. | Ensemble $\Delta$(%) |
|---|---|---|---|---|---|
| All Sources/(N=10) | $0.25 \pm 0.041$ | $0.0068 \pm 0.0010$ | 10.7 | 0.870 | 1.85 |
| All Sources/(N=25) | $0.26 \pm 0.034$ | $0.0073 \pm 0.0007$ | 10.7 | 0.873 | 1.86 |
| All Sources/(N=50) | $0.24 \pm 0.026$ | $0.0076 \pm 0.0007$ | 10.7 | 0.871 | 1.86 |
| All Sources/(N=75) | $0.25 \pm 0.021$ | $0.0078 \pm 0.0006$ | 10.7 | 0.872 | 1.87 |
| All Sources/(N=100) | $0.26 \pm 0.018$ | $0.0080 \pm 0.0005$ | 10.7 | 0.872 | 1.87 |
| Data Shuffling/(N=10) | $0.25 \pm 0.042$ | $0.0061 \pm 0.0010$ | 10.8 | 0.871 | 1.86 |
| Data Shuffling/(N=25) | $0.22 \pm 0.035$ | $0.0080 \pm 0.0009$ | 10.7 | 0.870 | 1.89 |
| Data Shuffling/(N=50) | $0.25 \pm 0.027$ | $0.0079 \pm 0.0007$ | 10.7 | 0.870 | 1.88 |
| Data Shuffling/(N=75) | $0.25 \pm 0.026$ | $0.0084 \pm 0.0006$ | 10.7 | 0.870 | 1.87 |
| Data Shuffling/(N=100) | $0.25 \pm 0.021$ | $0.0082 \pm 0.0005$ | 10.7 | 0.871 | 1.86 |
| Random Bit Change/(N=10) | $0.26 \pm 0.047$ | $0.0063 \pm 0.0011$ | 10.7 | 0.874 | 1.88 |
| Random Bit Change/(N=25) | $0.23 \pm 0.027$ | $0.0075 \pm 0.0009$ | 10.6 | 0.874 | 1.85 |
| Random Bit Change/(N=50) | $0.22 \pm 0.019$ | $0.0069 \pm 0.0006$ | 10.6 | 0.874 | 1.83 |
| Random Bit Change/(N=75) | $0.22 \pm 0.017$ | $0.0070 \pm 0.0005$ | 10.6 | 0.874 | 1.83 |
| Random Bit Change/(N=100) | $0.21 \pm 0.014$ | $0.0068 \pm 0.0004$ | 10.6 | 0.874 | 1.82 |

Table 8: No. of models v/s Variability. N denotes total number of trained models.

216

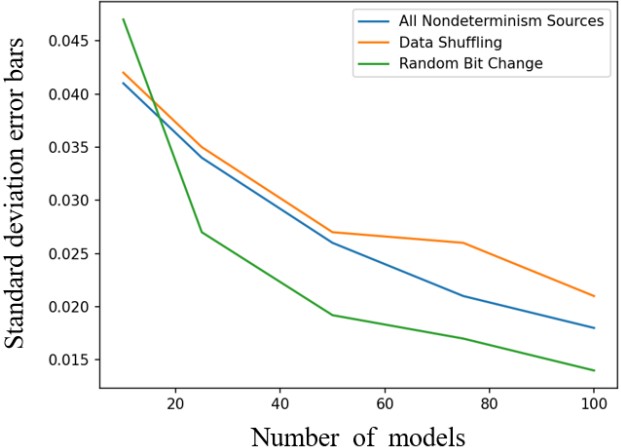

Figure 2: Change in error with respect to number of trained models

217  **Epochs v/s Variability**   Since we found that a smaller number of trained models is less likely to affect the result in
218  terms of model variability, we kept the number of trained models constant at 25. In addition to the original settings, we
219  changed the number of epochs from 100 to 500 to see its impact on model variability. It can be seen that changing the

epochs does not affect the performance variability, but the model diversity. The values for pairwise correlation and change in two models ensembling indicate greater model diversity as the number of epochs increases. The same trend can be observed in both sources, as shown in Table 9.

| Setting | Accuracy SD (%) | Cross-Entropy SD (%) | Pairwise Disagree (%) | Pairwise Corr. | Ensemble Δ(%) |
|---|---|---|---|---|---|
| Data Shuffling/100 | $0.20 \pm 0.02$ | $0.0056 \pm 0.0007$ | 10.7 | 0.910 | 1.64 |
| Data Shuffling/200 | $0.24 \pm 0.03$ | $0.0058 \pm 0.0009$ | 10.6 | 0.895 | 1.75 |
| Data Shuffling/300 | $0.23 \pm 0.02$ | $0.0070 \pm 0.0010$ | 10.5 | 0.884 | 1.79 |
| Data Shuffling/400 | $0.24 \pm 0.03$ | $0.0072 \pm 0.0011$ | 10.6 | 0.878 | 1.80 |
| Data Shuffling/500 | $0.22 \pm 0.03$ | $0.0080 \pm 0.0009$ | 10.7 | 0.870 | 1.89 |
| Random Bit Change/100 | $0.23 \pm 0.03$ | $0.0071 \pm 0.0013$ | 10.5 | 0.913 | 1.58 |
| Random Bit Change/200 | $0.22 \pm 0.02$ | $0.0076 \pm 0.0007$ | 10.3 | 0.898 | 1.71 |
| Random Bit Change/300 | $0.24 \pm 0.03$ | $0.0073 \pm 0.0012$ | 10.3 | 0.887 | 1.68 |
| Random Bit Change/400 | $0.26 \pm 0.03$ | $0.0076 \pm 0.0010$ | 10.5 | 0.881 | 1.83 |
| Random Bit Change/500 | $0.23 \pm 0.03$ | $0.0075 \pm 0.0009$ | 10.6 | 0.874 | 1.85 |

Table 9: No. of Epochs v/s Variability

**Generalization**   In this section, we examine the first two claims of the paper on a larger scale in terms of architecture and dataset. So far, these claims have shown no difference in results and have only been tested with ResNet-18 representing the largest architecture in [1]. For this reason, we went a step further and conducted experiments to test the generalization of nondeterminism and instability to CIFAR-100 (dataset) and ResNet-34 (model architecture). We conducted the experiment to obtain the 25 trained models with 200 epochs each for the sources of nondeterminism listed in Table 10. We obtained an average accuracy of 63% for the CIFAR-100 test dataset, however the goal is to observe the changes in model variability and its metrics. We have found that the two different sources of nondeterminism produce roughly the same variability. The relative variability of instability with "Random Bit Change" also shows a similar result. However, the significant change in values of these metrics are observed about three times higher than the experiments performed with CIFAR-10 and ResNet-14. Even though this difference is due to the lower accuracy of the test results, the two main claims still hold.

| Nondeterminism Sources | Accuracy SD (%) | Cross-Entropy SD (%) | Pairwise Disagree (%) | Pairwise Corr. | Ensemble Δ(%) |
|---|---|---|---|---|---|
| Parameter Initialization | $0.80 \pm 0.19$ | $0.029 \pm 0.005$ | 31.4 | 0.768 | 2.63 |
| Data Shuffling | $0.78 \pm 0.11$ | $0.037 \pm 0.005$ | 34 | 0.748 | 3.07 |
| Random Bit Change | $0.73 \pm 0.13$ | $0.040 \pm 0.009$ | 33.7 | 0.775 | 3.03 |

Table 10: Generalization of nondeterminism and instability

# 5   Discussion

Our results in section 4 fully support the first two assertions regarding the effects of "non-determinism" and its identified cause "instability". To gain sufficient confidence in the result, we also tested these claims on larger architectures and datasets, which also confirms the results of the study. But, when conducting experiments with accelerated ensembling and TTA as a solution to reduce variability, some differences were found. Our results show that both approaches can reduce variability. However, the extent to which they reduce variability is presented higher in the original study and lacks concrete numbers. In addition, we did not find in the paper a mathematical formulation for the average percentage of variability reduction that could have avoided this discrepancy. However, this is not sufficient to refute the claim. Therefore, we also support the third claim. Moreover, the discovery that all sources of nondeterminism have similar effects on model variability is novel in itself and opens many interesting areas of research toward reproducibility of deep neural networks.

**Strengths and weaknesses**   One of our strengths in the reproducibility study was that we stuck to the original implementation by using the publicly provided code and were able to create the experimental environment with exact

hardware and software specifications. This allowed us to obtain similar results that confirmed the paper's claims. Another strength of our work was to perform some additional experiments that helped us reduce the overall computation time, allowing experiments with a larger architecture and dataset to be completed on time. The weakness of our approach is that in the limited time available for the reproducibility study, we could not test the claims about different combinations of hyperparameters, since 100 of trained models must be seeded for each experiment, whereas training a single model takes about 40 minutes.

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
