# OpenReview forum: "[RE] Nondeterminism and Instability in Neural Network Optimization"
_ML_Reproducibility_Challenge/2021/Fall — RC2021_

### Official Review · Reviewer_pHPD · 2022-02-27
**Solid submission that uses official code release, but also goes beyond original work in a few ways.**

**Rating:** 7
**Confidence:** 4

**Review:**

The authors identified three key claims from the original paper, and using the original code with some slight modifications to run it on their system, they were able to support those three key claims.
They  found quantitative but not qualitatively different results  for the third claim,  regarding the effectiveness of Test time data augmentation and accelerated ensembling as mitigation techniques.  Specifically they found those techniques to be roughly 20% less effective than reported in the original paper.

The author is also performed three extra experiments.   The first two show that similar results to those of the original paper can be achieved using less compute: 10 instead of 100 models or training for fewer epochs.
The third experiment performs a similar but more limited set of experiments on CIFAR 100 with Resnet 34 and finds qualitatively similar results with larger variability between models.

Strengths:
- The submission is clearly written and clearly states the authors contributions.
- The replication followed good practice  across the board.
-  The extra experiments are valuable especially the CIFAR 100 experiments,  Which suggests that larger models and/or more complicated tasks lead to greater variability.

A few areas for improvement:
- A few points weren't entirely clear; for instance when the author's mentioned that they did not find an open source implementation of CKA they should also state that they did not include it in their experiments.
- I also think the submission would benefit from restating a few of the key ideas and techniques of the original paper.
- In my mind the most surprising result from the original paper is that changing is single trailing bit in the initialization produces as much variability as any other intervention.  I would've loved to see that result replicated from scratch, as it is so surprising.
-  It would have been interesting to try and disentangle the effect of changing the task versus changing the architecture.   I suspect the greater variability is due to using a deeper network but it would be great to confirm that experimentally.

---

### Official Review · Reviewer_wFzJ · 2022-02-28
**Detailed documentation on the reproduced results**

**Rating:** 7
**Confidence:** 4

**Review:**

The paper looks into the reproductivity of the paper "Nondeterminism and instability in neural network optimization" by Summers and Dinneen from ICML 2021.

Strengths:
- Details report on how reproducibility has been verified, including potential challenges and caveats.
	- Additional ablation and generalization experiments.

Weaknesses:
	- The evaluation focuses on just the computer vision tasks, whereas the original paper studied both computer vision and language modeling.

Overall, it is a pretty good reproducibility report, where the authors document clearly how they reproduce the results, including details in the code, model choice, datasets, hyperparameters, and computational requirements. It also documents clearly some of the challenges (e.g., training cost) and caveats (not exactly matched variability reduction ratio).

In more detail, the paper focuses on examining three main claims from the original paper.  First, it performs a quantitive evaluation of the impact of each source of nondeterminism in NN optimizations using ResNet-14/CIFAR-10 and shows that all of them have a similar effect on model diversity and variability. Second, it verifies that any source of nondeterminism is susceptible to tiny changes to the initial parameters. Third, it verifies the two methods proposed by the original paper are able to reduce model variability without additional training cost.  All three claims appear to have been verified.

The reviewer also appreciates the authors for conducting additional experiments including an ablation study that shows the impact of reducing the training epochs and generalization experiments on larger architectures and datasets. Both are useful for future research.

---

### Official Review · Reviewer_U2wX · 2022-03-06
**Interesting experiments and clear writing**

**Rating:** 8
**Confidence:** 3

**Review:**

Summary
The authors reproduce the results for the paper “Nondeterminism and Instability in Neural Network Optimization” and use the implementation provided by the original authors. They have contacted the original authors, and discussed the experimental setting. The scope of reproducibility is clearly stated and supported in the rest of the paper. The authors have provided extended experiments to ease replicating the results in terms of the computational time and resources.


Pros

•	The paper is well structured and organized. The flow of information is very good and overall the paper was pleasure to read.

•	The authors provide two extensions to the original paper. They study the effect of a number of models and training epochs in the variability reduction results.

•	Authors have provided a good discussion on the results. They have explained a source of possible computation error in the original implementation due to the gap in the results for verifying claim 3.

Cons

•	As the authors also mentioned as a downside of this report, it would be nice to verify the results for the extended experiments also for the other nondeterminism sources.
Minor comments

•	Table 1 is not self-contained and is a bit difficult to get the information without going through the text. So, I suggest to add more description about the table’s format in its caption.


The overall structure and quality of the writing is very good. The extended experiments seems interesting since it makes the reproducibility of results in the environments with limited resources feasible. Therefore, I recommend acceptance.

---

### Meta-Review · Area_Chair_bn1n · 2022-04-07

**Recommendation:** Accept
**Confidence:** 5

**Metareview:**

All reviewers strongly agree on the high quality of reproduction effort in this report. The reproduction effort is heavy with experiments, and the authors do a good job in performing ablation studies and generalization experiments on different dataset and model architectures. The report adds valuable insights to the results of the original paper, and hence I recommend for acceptance.

---

### Decision · Program_Chairs · 2022-04-09

**Decision:**

Accept

**Comment:**

Following the recommendation of reviewers and meta-reviewer, the paper is accepted for ML Reproducibility Challenge 2021, and will be published in the upcoming special edition of ReScience Journal.